# Discrimination and Health: The Mediating Effect of Acculturative Stress

**DOI:** 10.3390/ijerph18105312

**Published:** 2021-05-17

**Authors:** Alfonso Urzúa, Alejandra Caqueo-Urízar, Diego Henríquez, David R. Williams

**Affiliations:** 1Escuela de Psicología, Universidad Católica del Norte, Antofagasta 1240000, Chile; 2Instituto de Alta Investigación, Universidad de Tarapacá, Arica 1000000, Chile; acaqueo@academicos.uta.cl (A.C.-U.); xdiegohenriquez@gmail.com (D.H.); 3TH Chan School of Public Health, Harvard University, Boston, MA 02115, USA; dwilliam@hsph.harvard.edu

**Keywords:** migration, acculturative stress, discrimination, racism, health, mental health

## Abstract

There is not much evidence on the effects of south–south migration and its consequences on physical and mental health. Our objective was to examine the mediating role of Acculturative Stress in the association between ethnic discrimination and racial discrimination with physical and mental health. This research is a non-experimental, analytical, cross-sectional study. A total of 976 adult Colombian migrants living in Chile were interviewed. We used the Everyday Discrimination Scale, the acculturative stress scale, and the Medical Outcomes Study Short Form (SF-12) for health status; we found that racial and ethnic discrimination had a negative effect on physical and mental health. In the simultaneous presence of both types of discrimination, racial discrimination was completely absorbed by ethnic discrimination, the latter becoming a total mediator of the effect of racial discrimination on mental and physical health. Our findings are consistent with the literature, which suggests that there are various types of discrimination which, individually or in their intersectionality, can have negative effects on health.

## 1. Introduction

In 2019, 3.5 percent of the world’s population, about 272 million people, were international migrants [1]. Migrating produces effects both on the person who migrates and on the society that receives them, derived mainly from factors linked to the interaction between the two. Two of these factors are perceived as discrimination and acculturative stress, variables that can directly or indirectly affect the physical and mental health of the migrant [2].

Discrimination is conceptualized as the different treatment of a group with common characteristics or of a person belonging to such a group [3]. The perception of being a target of discrimination (isolation and unfair treatment) has been adversely related to both physical and mental health. This pattern has been consistently reported for various types of discrimination and for multiple populations [4,5,6,7,8,9,10,11,12,13,14,15].

In migrant populations, two of the most studied types of discrimination are those based on ethnicity and race, which have been associated with poorer physical and mental health [16,17,18,19,20,21,22,23,24], low self-esteem [25,26], and reduced well-being [27,28,29,30,31,32,33].

The link between discrimination and poor health can be explained by theories about the ways in which stress can affect health [9,34,35]. One particular type of stress is a result of the process of acculturation, understood as the phenomenon derived from continuous and direct contact between groups of people from different cultures, which can lead to changes in the cultural patterns of one or both groups [36]. At the psychological level, it can imply dealing with demands as one confronts new attitudes and values, the need to acquire new social skills and norms, and changes in one’s own group identity and the adjustment or adaptation to a different environment [37]. When these demands of adaptation to a new culture exceed people’s capacities to face them, acculturative stress can arise [38,39,40].

Acculturative stress has been linked to the presence of anxiety and depressive disorders, as well as feelings of marginality, increased psychosomatic disorders, eating behavior, and identity confusion [41,42,43,44,45,46,47,48]. It can also have a mediating effect on the relationship between discrimination and health [9], acculturation and sedentarism [49], the association between acculturation and psychological functioning [50], or between discrimination and psychological stress [51].

Although the literature on the relationship between discrimination and physical and mental health is extensive, research on factors that mediate and/or moderate this relationship in migrant populations is scarce, especially in Latin America. Our review has not found references to studies conducted with South American migrants in a South American country, nor any that refer to the effects that acculturative stress could have, as a specific stressor faced by migrant populations, on the relationship between discrimination and health.

This research is contextualized in the so-called south–south migration processes, i.e., South Americans migrating to countries in the region, a phenomenon that has not been widely studied. In this study, we address Colombian migration to Chile, which is mostly of African descent and which, given its upsurge in recent times, has generated situations of social tension, such as discrimination against migrants, either racially or because of their country of origin. These situations are linked in the social imaginary to drug trafficking, drugs and, in the case of women, sex trade [52,53]. In this context, this study aimed to examine the mediating role of acculturative stress in the association between ethnic discrimination and racial discrimination with physical and mental health. We hypothesize that stress will have a mediating effect on this relationship.

## 2. Materials and Methods

### 2.1. Sample

This research is a non-experimental, analytical, cross-sectional study. Given the challenges of identifying the target population, non-probability sampling methods were used that combined snowball sampling techniques with purposive sampling for hard-to-reach groups. Inclusion criteria were a Colombian resident in Chile (not a tourist), over 18 years of age and having been in Chile for more than 6 months. No exclusion criteria were applied. The participants were recruited in three of the cities with the highest rates of Colombian migrants in the country. They were interviewed mainly in public institutions, such as the Chilean Catholic Migration Institute (INCAMI), Global Citizen-Jesuit Migrant Services, Department of Immigration, Colombian Consulate, their workplaces, health centers, and neighborhoods with high migrant populations, among others.

### 2.2. Measures

To identify respondents’ sociodemographic data, questions were asked relating to years of age, biological sex, nationality, level of studies (e.g., no education, primary school, secondary school, university), years of permanence in the country, city of residence, type of residence, monthly income, and work situation (without work, self-employed, dependent worker, etc.). All the questionnaires described below were previously assessed in Latin American migrant populations through cognitive interviews. These have been used in previous studies on South American migrants.

### 2.3. Perceived Discrimination

We used the Spanish version of the Everyday Discrimination Scale [54,55]. The questions asked participants to indicate how often they had been treated badly or unfairly in 10 situations. We asked respondents separately to answer the questions on this scale by thinking first about their skin color and then considering their nationality. In the present research, the scale for ethnic discrimination had an Alpha of 0.91, while the scale for racial discrimination had an Alpha of 0.93. We have assessed ethnic discrimination separately from racial discrimination, as there is evidence from previous studies that the two types of discrimination function differently [31].

### 2.4. Health Status

Health was assessed using the Medical Outcomes Study Short Form—SF-12 [56], adapted to Spanish by Vilagut et al. [57]. It consists of 12 items grouped into two dimensions, one for mental health and the other for physical health [57,58]. There is evidence of the reliability of this scale for use in the Colombian population [59].

### 2.5. Acculturative Stress Scale

We used a brief version of the acculturative stress scale developed by Ugalde-Watson et al. [60] to evaluate the levels of stress experienced during the acculturation process. This scale consists of 14 items grouped into three dimensions: (a) the stress derived from the preparation and departure from the country of origin, (b) the stress produced by socioeconomic concerns in the host country, and (c) the tensions typical of adaptation to sociocultural changes or Chilean society [61].

### 2.6. Procedures

The present research is part of the FONDECYT 1180315, which has been reviewed and approved by the Scientific Ethics Committee of Universidad Católica del Norte, Chile. Each of the participants signed a consent form. The data were coded and analyzed with the SPSS-21 software (IBM Corp., Armonk, NY, USA) [62], while the Mplus 8.2 software (Muthen & Muthen, Los Angeles, CA, USA) was used for structural model adjustment analyses [63].

### 2.7. Statistical Analysis

First, the measurement models for each of the scales were estimated by means of confirmatory factor analyses. Subsequently, four structural equation models were examined (M1, M2, M3 and M4). The first model (M1) estimated the effect of ethnic discrimination and racial discrimination on physical and mental health. The second model (M2) estimated the effects of ethnic and racial discrimination, mediated by acculturative stress, on physical and mental health. The third model (M3) estimated the effect of racial discrimination on mental and physical health. Finally, the indirect effect of racial discrimination on physical and mental health was estimated and mediated in a serial manner by ethnic discrimination and acculturative stress.

The indirect effects of the mediation model (M2, M4) were estimated following the recommendations of Stride, Gardner, Catley, and Thomas [64]. The models were analyzed with the Mplus 8.2 software [58], using the robust weighted least squares (WLSMV) estimation method, which is robust for non-normal ordinal variables [65]. The goodness-of-fit values of all structural models were estimated using Chi-square values (χ2), the approximation mean square error (RMSEA), the comparative fit index (CFI), and the Tucker–Lewis index (TLI). According to recommended standards in the literature (e.g., Schreiber, 2017), RMSEA ≤ 0.08, CFI ≥ 0.95, and TLI ≥ 0.95 values are considered adequate and indicative of a good fit.

Analyses were performed controlling for city, sex and age, and no significant differences were found in the reported values due to the effect of these variables.

## 3. Results

### 3.1. Participants

A total of 976 adult Colombian migrants were interviewed, of which 499 (51.1%) resided in Antofagasta, 250 (25.6%) in Arica, and 227 (23.3%) in Santiago. The sample was almost evenly divided by sex with 477 (48.9%) males and 496 (50.8%) females, with missing data for 3 (0.3%). The age of participants ranged from 18 to 89 years (ME = 35.43; SD = 10.28).

### 3.2. Descriptives

Table 1 shows the average values, as well as the range of scores, for each of the variables studied in the model.

### 3.3. Measurement Models

Table 2 shows the goodness-of-fit indices of the estimated measurement models. The estimation of the CFI and TLI indicators were adequate in all models, being close to the expected values (CFI > 0.95; TLI > 0.95). However, the estimated RMSEA in all variables exceeded the recommended cut-off point (RMSEA > 0.08) [66]. In this scenario, the results of the structural models should be interpreted with caution, considering this limitation.

Table 3 shows the goodness-of-fit indices of the estimated structural models. All the models presented indicators close to the criteria recommended by the literature, and therefore are good representations of the observed relationships.

In M1, we observe that ethnic discrimination exerts a moderate inverse effect (b > 0.30) [67] on physical and mental health. No significant effects of racial discrimination on physical and mental health are observed (see Figure 1).

In M2, we observe that acculturative stress completely mediates [68] the relationship between ethnic discrimination and physical and mental health (see Figure 2). In contrast, there was no evidence to support the of the idea that acculturative stress mediates the relationship between racial discrimination and physical and mental health.

With regards to direct effects, ethnic discrimination had a positive effect of small magnitude (b > 0.20) [67] on AS, and inverse effects of small magnitude (b > 0.20) on physical and mental health. The racial discrimination did not show statistically significant direct effects on acculturative stress, physical or mental health.

Since we did not detect any significant effect of racial discrimination on the other variables, we generated the new hypothesis that ethnic discrimination could be mediating the relationship between racial discrimination and physical and mental health. Therefore, we tested two alternative models (M3 and M4). The first (M3) estimated the effect that racial discrimination has on physical and mental health, while the second one (M4), estimated a serial mediation model, where ethnic discrimination followed by acculturative stress acted as mediators of the relationship between racial discrimination and physical and mental health.

As can be seen in Figure 3 (M3), racial discrimination showed small inverse effects (b > 0.20) on physical and mental health. Therefore, this gives us a signal to think that, in the previous models, ethnic discrimination may be mediating the relationship between racial discrimination and physical and mental health.

As can be seen in Figure 4, the racial discrimination did not have statistically significant direct effects on acculturative stress, physical and mental health. However, it does show a large positive effect (b > 0.50) [67] on ethnic discrimination. On the other hand, ethnic discrimination has a small positive effect (b > 0.20) on acculturative stress, and inverse effects on physical and mental health. In the case of acculturative stress, it has a moderate inverse effect (b > 0.30) on physical and mental health.

Concerning indirect effects, M4 shows that ethnic discrimination completely mediates the relationship between racial discrimination and physical and mental health (see Figure 4). Acculturative stress does not have significant indirect effects on the relationship between racial discrimination and physical and mental health. Finally, ethnic discrimination followed by acculturative stress completely mediates the relationship between physical and mental health.

## 4. Discussion

The data analyzed in our research allows us to highlight two results that we believe are important. Firstly, our findings are consistent with the literature, which suggests that there are various types of discrimination which, individually or in their intersectionality, can have negative effects on health. The results reported here reinforce the earlier evidence, since both racial and ethnic discrimination had a negative effect on physical and mental health. In addition, our analyses contribute to the idea that, in the face of the simultaneous presence of both types of discrimination, racial discrimination was completely absorbed by ethnic discrimination, the latter becoming a total mediator of the effect of racial discrimination on mental and physical health. Similar effects have been reported by Zainiddinov [69], who reports in a study with different Muslim ethnic groups that, despite a common basis of discrimination (being Muslim), ethnicity played the more important role in perceived discrimination, supporting the hypothesis that people with multiple intersections and disadvantaged statuses are discriminated against at higher rates than those with only one stigmatized status.

We believe that the effect caused by belonging to an ethnic category (in this case being Colombian) may have various explanations at the base that arise due to recent migration, these being the maintenance of a strong ethnic identity and the widespread presence of negative stereotypes of this group. The Colombian migration into Chile is a relatively recent migration (last few years), that is unlike earlier historical migrations from countries in South America, such as Peru. Peruvian migrants do not manifest a strategy of acculturation associated with integration or biculturalism, but try to maintain their own ethnic identity, sometimes exacerbating it, which can cause greater discrimination from the Chilean population that perceives them as a threat to the dominant culture. In addition, it has been found that, in countries with a recent migratory flow, such as this one, native people tend to prefer that newcomers adopt the practices and customs of the host country, i.e., assimilate into the dominant culture [70].

The maintenance of this strong identity is reinforced by less contact with the host population and marked differences in relational style, which often makes Columbian immigrants function as a group separately from the host society, thus becoming more socially visible by remaining a different group with little permeability to the practices, customs and lifestyles of the host population, thereby generating more behaviors of self-exclusion when they perceive themselves to be discriminated against [71].

The recent nature of this migration also interacts with our second hypothesis, concerning the prevalence of negative stereotypes about the Colombian migrant. Colombian migration is linked to the perceived transfer of violence from the country of origin, which often caused migration combined with an exacerbated sexualization in the case of women, generating greater discrimination from the host population because they are Colombian, and not so much because of the color of their skin. Echeverri [72] suggests that this is because of social representations built around Colombian migration, associated with violence and unwanted stigmatized subjects. In this context, being Colombian and the prejudices and stereotypes associated with it not only increases racial discrimination but is also strong enough that at least some of these stereotypes are accepted as true by Columbian migrants themselves.

Secondly, we found that acculturative stress had a mediating effect on the relationship between discrimination and health. Traditionally, the focus of many studies of stress have been on the inverse relationship it has with various indicators of health and well-being [33,38,73,74]. Few studies have considered it a mediating variable [9,49,50,51]. In this context, our results provide evidence that stress can mediate the relationship between ethnic discrimination and its effect on physical and mental health, probably because this is one of the main outcomes of the discrimination process, and this in turn has a direct effect on the increase of physical and mental health problems.

As has been mentioned, although the literature on the relationship between discrimination and physical and mental health or between discrimination and well-being is extensive, research on factors that mediate and/or moderate this relationship in the migrant population, and especially in Latin America, is still incipient. Accordingly, this research provides new insight regarding how psychosocial factors can affect physical and mental health in the context of these new mass -migratory movements of South American immigrant populations in the Southern Cone of the Americas. Given the frequent exposure of different ethnic groups to discrimination and the high impact of discrimination on individual suffering, loss of well-being and physical and mental health problems, it is important for future research to continue to examine the relationship between discrimination and health (physical and mental) and well-being, including factors that influence the strength of this relationship [74].

The research findings here must be considered in the light of its limitations. First, given the cross-sectional design used, it was not possible to examine the associations between discrimination and health—either physical or mental—over time or to examine the temporal ordering of the associations. Second, given the sampling strategies used, we cannot have confidence that our findings are representative of all Columbian immigrants. Third, the findings from this study cannot be generalized to all South American immigrants to Chile. It is necessary for future research to evaluate how the analyzed models behave in different migrant populations, particularly due to the impact that prejudice seems to have in Chile for certain ethnic groups and the type of acculturation strategy that the migrating population has adopted. In this context, it is necessary to deepen our understanding on the role of culture in the way acculturative stress can influence particular immigrant populations [75].

Possible lines of research to be followed should explore the role that ethnic and racial identity, the search for social support, and the management of emotions have as possible coping strategies for acculturative stress [76], given that such research can provide inputs that allow for clarifying the correlates of discrimination and the needed target factors, so that interventions would be more effective.

## 5. Conclusions

Racial and ethnic discrimination had a negative effect on physical and mental health. In the simultaneous presence of both types of discrimination, racial discrimination was completely absorbed by ethnic discrimination, with the latter becoming a total mediator of the effect of racial discrimination on mental and physical health. These findings provide evidence to policy makers of the need to intervene at the societal level and in terms of social imaginaries and stereotypes about migrants associated with their ethnic origin, especially in public discourse and policies, generating a positive image of migrants, thus contributing to less discrimination towards this population.

## Figures and Tables

**Figure 1 ijerph-18-05312-f001:**
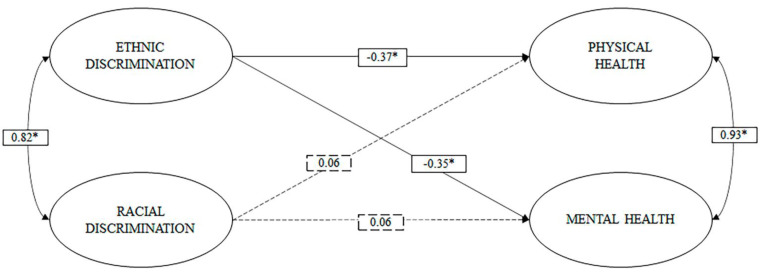
Model of direct effects of ethnic and racial discrimination on physical and mental health. Note: * *p* < 0.05; dashed lines represent non-significant routes.

**Figure 2 ijerph-18-05312-f002:**
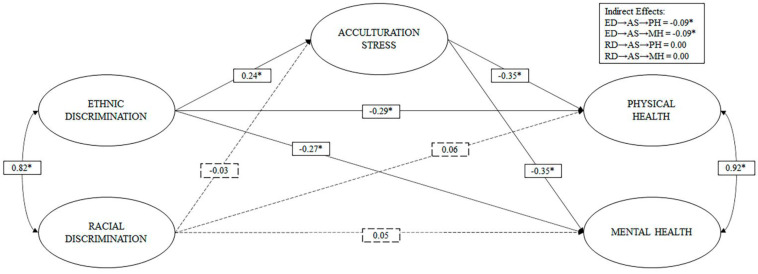
Model of indirect effects of acculturative stress on the relationship between ethnic and racial discrimination and physical and mental health. Note: * *p* < 0.05; dashed lines represent non-significant routes; ED, ethnic discrimination; RD, racial discrimination; AS, acculturative stress; PH, physical health; MH, mental health.

**Figure 3 ijerph-18-05312-f003:**
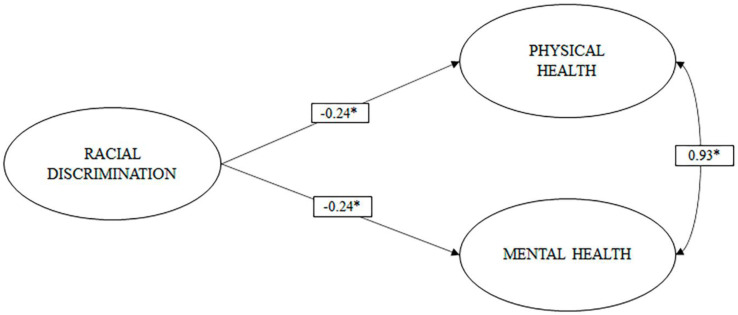
Model of direct effects of racial discrimination on physical and mental health.

**Figure 4 ijerph-18-05312-f004:**
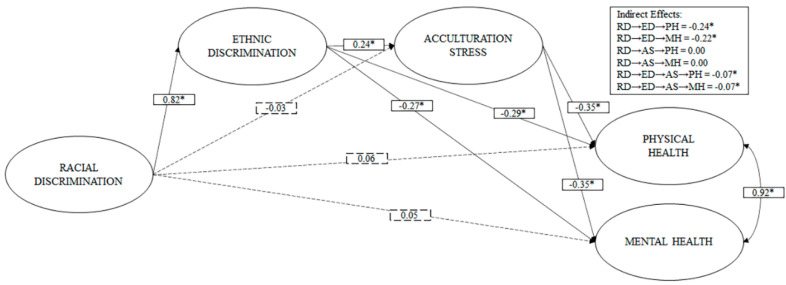
Model of direct effects of racial discrimination, mediated by ethnic discrimination and acculturative stress, on physical and mental health. Note: * *p* < 0.05; dashed lines represent non-significant routes; ED, ethnic discrimination; RD, racial discrimination; AS, acculturative stress; PH, physical health; MH, mental health.

**Table 1 ijerph-18-05312-t001:** Descriptive statistics.

Variables	Mean	SD	Range	Min	Max	*n*
Perceived discrimination						
Racial discrimination	0.55	0.70	3	0	3	837
Ethnic discrimination	0.72	0.71	3	0	3	930
Acculturation stress	3.07	0.91	4	1	5	744
Health status						
Physical health	75.29	17.88	79	21	100	942
Mental health	70.92	17.42	83	17	100	953

Note: Health status scores were recoded in a range from 0 to 100.

**Table 2 ijerph-18-05312-t002:** Indicators of global adjustment of measurement models and structural models.

Models	Parameters	χ^2^	DF	*p*	CFI	TLI	RMSEA	RMSEA CI 90%	
								Low	Superior
ED	40	783.767	35	0.00	0.943	0.927	0.148	0.140	0.158
RD	40	691.463	35	0.00	0.965	0.956	0.146	0.137	0.156
AS	73	544.257	74	0.00	0.980	0.975	0.081	0.075	0.087
SF-12	52	408.221	19	0.00	0.949	0.925	0.145	0.133	0.157
M1	137	2436.596	344	0.00	0.940	0.934	0.079	0.076	0.082
M2	214	3012.959	806	0.00	0.954	0.951	0.053	0.051	0.055

ED, ethnic discrimination; RD, racial discrimination; AS, acculturative stress; SF-12, physical and mental health.

**Table 3 ijerph-18-05312-t003:** Indicators of global adjustment of structural models.

Models	Parameters	χ^2^	DF	*p*	CFI	TLI	RMSEA	RMSEA CI 90%	
								Low	Superior
M1	137	2436.596	344	0.00	0.940	0.934	0.079	0.076	0.082
M2	214	3012.959	806	0.00	0.954	0.951	0.053	0.051	0.055
M3	94	831.929	132	0.00	0.970	0.966	0.074	0.069	0.079
M4	214	3012.959	806	0.00	0.954	0.951	0.053	0.051	0.055

CFI, comparative adjustment index; TLI, Tucker–Lewis index; RMSEA, root mean square error of approximation.

## Data Availability

The data presented in this study are available on request from the corresponding author. The data are not publicly available because the project has state funding and will only be released once the project is finished.

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
