# Peer review of "Discrimination and Health: The Mediating Effect of Acculturative Stress"

_ijerph, 2021, doi:10.3390/ijerph18105312_

Round 1

Reviewer 1 Report

Reviewer comments:

Thank you for providing me the opportunity to review this manuscript. It is an important contribution to the field, as studies on the psychological effects of discrimination in a variety of fields is still lacking. The paper has some shortcomings that need to be addressed though, before being considered for publication.

The introductory sentence is a bit unnecessary to my taste – I would assume that the reader knows what international migration is. The last sentence of the first paragraph needs a reference.

There are a number of typos/spelling mistakes, for example row 34 (omit “a” before “common characteristics”). Also, the language especially in the introduction is at times cumbersome.

Use the same terminology throughout, for example you use “acculturation stress” on row 48, but “acculturative stress” on row 56.

Sample: Please give more information about the sample – how were they found and recruited, can they be considered representative of the population?

Measures: Please give more detail about the measures, including the answer categories that were possible for each measure (age, sex, nationality etc)

Have the EDS and acculturation stress scale that were used been validated in your setting?

Results: Please provide background descriptive data on your sample before going into presenting the models.

Is the assessment of the effect of racial discrimination of ethnic discrimination really worthwhile? You are assessing the effects of very closely interlinked concepts on each other and inferring causality, which to my understanding is not theoretically plausible.

Author Response

  1. Thank you for providing me the opportunity to review this manuscript. It is an important contribution to the field, as studies on the psychological effects of discrimination in a variety of fields is still lacking. The paper has some shortcomings that need to be addressed though, before being considered for publication.

Response: Thank you very much for your review. We have incorporated all of your suggestions into the manuscript.

  1. The introductory sentence is a bit unnecessary to my taste – I would assume that the reader knows what international migration is. The last sentence of the first paragraph needs a reference.

Response: We have replaced the first sentence with a specific piece of information that highlights the importance of the phenomenon under study.   We have added the required citation at the end of the sentence.

In 2019, 3.5 per cent of the world's population, about 272 million people, were international migrants. [1]. Migrating produces effects both on the person who migrates and on the society that receives them, derived mainly from factors linked to the interaction between the two. Two of these factors are perceived discrimination and acculturation stress, variables that can directly or indirectly affect the physical and mental health of the migrant [2]

  1. There are a number of typos/spelling mistakes, for example row 34 (omit “a” before “common characteristics”). Also, the language especially in the introduction is at times cumbersome.

Response: we have corrected the wording of the requested sentence and revised the wording again

  1. Use the same terminology throughout, for example you use “acculturation stress” on row 48, but “acculturative stress” on row 56.

Response: we have chosen to use the concept of "acculturative stress" and have corrected it throughout the manuscript.

  1. Sample: Please give more information about the sample – how were they found and recruited, can they be considered representative of the population?

Response: Information requested has been added.

This research is a non-experimental, analytical, cross-sectional study.  Given the challenges of identifying the target population, non-probability sampling methods were used that combined snowball sampling techniques with purposive sampling for hard-to-reach groupsThe participants were recruited in three of the cities with the highest rates of Colombian migrants in the country. They were interviewed in their workplaces, health centres, government agencies such as the migration department, neighbourhoods with high migrant populations, among others.

  1. Measures: Please give more detail about the measures, including the answer categories that were possible for each measure (age, sex, nationality etc)

Response: Information requested has been added.

To identify respondents’ sociodemographic data, questions were asked relating to years of age , biological sex, nationality, level of studies (ex: no education, primary school, secondary school, university), years of permanence in the country, city of residence, type of residence, monthly income, and work situation (without work, self-employed, dependent worker, etc.).

  1. Have the EDS and acculturation stress scale that were used been validated in your setting?

Response: We have added:

All the questionnaires described below were previously assessed in Latin American migrant populations through cognitive interviews.  These have been previously used in previous studies on South American migrants.

  1. Results: Please provide background descriptive data on your sample before going into presenting the models.

Response: We have added in the text

Variables

Mean

SD

Range

Min

Max

n

Perceived Discrimination
     Racial discrimination

     Ethnic discrimination

.55

.72

.70

.71

3

3

0

0

3

3

837

930

Acculturation stress

3.07

.91

4

1

5

744

Health status

     Physical health

     Mental health

75.29

70.92

17.88

17.42

79

83

21

17

100

100

942

953

  1. Is the assessment of the effect of racial discrimination of ethnic discrimination really worthwhile? You are assessing the effects of very closely interlinked concepts on each other and inferring causality, which to my understanding is not theoretically plausible.

Response: we have added the following paragraph in the section on instruments when describing the scale of discrimination.

We have assessed ethnic discrimination separately from racial discrimination, as there is evidence from previous studies that the two types of discrimination function differently [31].

Reviewer 2 Report

Very important study, especially as data on South American migrants are few. Good methodology, well written. The statistics look fine, but stats review might be good, as I am not specialised enough for that.

Author Response

  1. Very important study, especially as data on South American migrants are few. Good methodology, well written. The statistics look fine, but stats review might be good, as I am not specialised enough for that.

Response: Thank you very much for your review

Reviewer 3 Report

The authors aim to study the mediating role of acculturative stress in the association between ethnic discrimination and racial discrimination with physical and mental health. This is a very interesting topic. In general, this is a potentially stimulating article. I have some suggestions for your manuscript.

Introduction:

Your introduction needs to better inform the reader about the context of Chilean migration and politics of integration to contextualize the importance of adding specific knowledge about South American migrants in a South American country.

Your introduction, methods, results and discussion would benefit from being informed by intersectionality lens. This approach is missing.

Methods:

More detail concerning the data collection process is needed, namely to clarify recruitment process and eligibility criteria (inclusion and exclusion of participants).

Were ethnicity data collected?

Were all the scales used validated to Spanish?

Results:

Participants’ characterization should be presented in detail in the results section (and not in the methods section), and also results of the outcome measures and its correlation with participants’ characterization are needed to enable a better interpretation of results.

Discussion:

Participants’ characterization would enable a better interpretation and discussion of results.

The authors discuss a very interesting and relevant hypothesis (lines 210-211) “that people with multiple intersections and disadvantaged status are discriminated against at higher rates than those with only one stigmatized status”. Also, the double gender and ethnic discrimination (lines 231-240) would be very interesting to discuss based on the collected data. However results were not presented in a way to support this discussion.

Conclusion:

Finally, implications of the results for public policy are missing.

Author Response

  1. The authors aim to study the mediating role of acculturative stress in the association between ethnic discrimination and racial discrimination with physical and mental health. This is a very interesting topic. In general, this is a potentially stimulating article. I have some suggestions for your manuscript.

Response: Thank you very much for your review. We have incorporated all of your suggestions into the manuscript.

  1. Your introduction needs to better inform the reader about the context of Chilean migration and politics of integration to contextualize the importance of adding specific knowledge about South American migrants in a South American country. Your introduction, methods, results and discussion would benefit from being informed by intersectionality lens. This approach is missing.

Response: We have added a specific piece of information that highlights the importance of the phenomenon under study.  

This research is contextualised in the so-called south-south migration processes, i.e. South Americans migrating to countries in the region, a phenomenon that has not been widely studied.  In this study we address Colombian migration to Chile, which is mostly of African descent and which, given its massiveness in a short time, has generated situations of social tension such as discrimination against them, either racially or because of their country of origin, linked in the social imaginary to drug trafficking, drugs and sex trade in the case of women [52,53].

  1. Methods: More detail concerning the data collection process is needed, namely to clarify recruitment process and eligibility criteria (inclusion and exclusion of participants).

Response: We have added in the text

This research is a non-experimental, analytical, cross-sectional study.  Given the challenges of identifying the target population, non-probability sampling methods were used that combined snowball sampling techniques with purposive sampling for hard-to-reach groups.  Inclusion criteria were a Colombian resident in Chile (not a tourist), over 18 years of age and having been in Chile for more than 6 months.  No exclusion criteria were applied. The participants were recruited in three of the cities with the highest rates of Colombian migrants in the country. They were interviewed mainly in public institutions such as the Chilean Catholic Migration Institute (INCAMI), Global Citizen-Jesuit Migrant Services, Department of Immigration, Colombian Consulate, their workplaces, health centers, neighbourhoods with high migrant populations, among others.

  1. Were ethnicity data collected?

Response: Yes, was part of the inclusion criteria.

  1. Were all the scales used validated to Spanish?

Response: Yes, we have added this information in the description of each of the instruments used

  1. Results: Participants’ characterization should be presented in detail in the results section (and not in the methods section), and also results of the outcome measures and its correlation with participants’ characterization are needed to enable a better interpretation of results. Participants’ characterization would enable a better interpretation and discussion of results.

Response: we have added the requested table with the descriptives.  However, given that the relationship or differences in means between the different socio-demographic variables is far from the objective of this manuscript, we have decided to accept the reviewer's suggestion and have controlled the analysis by the most relevant variables.  In addition, the following paragraph has been added to the analysis section:

Analyses were performed controlling for city, sex and age, and no significant differences were found in the reported values due to the effect of these variables.

and this table was added in the results:

Variables

Mean

SD

Range

Min

Max

n

Perceived Discrimination
     Racial discrimination

     Ethnic discrimination

.55

.72

.70

.71

3

3

0

0

3

3

837

930

Acculturation stress

3.07

.91

4

1

5

744

Health status

     Physical health

     Mental health

75.29

70.92

17.88

17.42

79

83

21

17

100

100

942

953

  1. The authors discuss a very interesting and relevant hypothesis (lines 210-211) “that people with multiple intersections and disadvantaged status are discriminated against at higher rates than those with only one stigmatized status”. Also, the double gender and ethnic discrimination (lines 231-240) would be very interesting to discuss based on the collected data. However results were not presented in a way to support this discussion.

Response: Our assertion is based on direct work with migrant populations.  It is not the same to be a Dutch African descent as to be a Colombian.   It is not the same to be French as to be Colombian. We know that a migrant is likely to be doubly discriminated against either because of his or her skin colour or ethnic origin (depending on which prejudice weighs more heavily on the person who discriminates), but when both possibilities are present, they have a greater effect, even though our data show that when faced with both types of discrimination, ethnic discrimination is the one that prevails.   We would not know how else to explain it.

  1. Conclusion: Finally, implications of the results for public policy are missing.

Response: we have added in conclusions:  

These findings provide evidence to policy makers of the need to intervene at the societal level and in terms of social imaginaries and stereotypes about migrants associated with their ethnic origin, especially in public discourse and policies, generating a positive image of migrants, thus contributing to less discrimination towards this population.